# Thor: Wielding Hammers to Integrate Language Models and Automated Theorem Provers

**Albert Q. Jiang**
University of Cambridge
qj213@cam.ac.uk

**Wenda Li**
University of Cambridge
wl302@cam.ac.uk

**Szymon Tworkowski**
University of Warsaw
szy.tworkowski@gmail.com

**Konrad Czechowski**
University of Warsaw
konrad.czechowski@gmail.com

**Tomasz Odrzygóźdź**
IDEAS NCBR
tomaszo@impan.pl

**Piotr Miłoś**
Polish Academy of Sciences
pmilos@mimuw.edu.pl

**Yuhuai Wu**
Google Research & Stanford University
yuhuai@google.com

**Mateja Jamnik**
University of Cambridge
mateja.jamnik@cl.cam.ac.uk

## Abstract

In theorem proving, the task of selecting useful premises from a large library to unlock the proof of a given conjecture is crucially important. This presents a challenge for all theorem provers, especially the ones based on language models, due to their relative inability to reason over huge volumes of premises in text form. This paper introduces Thor, a framework integrating language models and automated theorem provers to overcome this difficulty. In Thor, a class of methods called hammers that leverage the power of automated theorem provers are used for premise selection, while all other tasks are designated to language models. Thor increases a language model's success rate on the PISA dataset from $39\%$ to $57\%$, while solving $8.2\%$ of problems neither language models nor automated theorem provers are able to solve on their own. Furthermore, with a significantly smaller computational budget, Thor can achieve a success rate on the MiniF2F dataset that is on par with the best existing methods. Thor can be instantiated for the majority of popular interactive theorem provers via a straightforward protocol we provide.

## 1 Introduction

In theorem proving, premise selection is the task of identifying useful facts from a large library that enable finding the proof of a given conjecture. It is essential for the discovery of many proofs, and Automated Reasoning in Large Theories (ARLT) depends on having apt methods for premise selection [Kühlwein et al., 2012, Sutcliffe et al., 2007]. A group of proof methods have been developed inside interactive theorem provers (ITPs) to deal with this task. They use external automated theorem provers (ATPs) to reach the remaining goals, inspect the proofs produced, and pick out the premises involved in them. Such systems are called hammers [Blanchette et al., 2016]. Hammers are available in many ITPs [Paulson, 2010, Kaliszyk and Urban, 2015, Gauthier and Kaliszyk, 2015, Czajka and Kaliszyk, 2018] and are immensely popular within the theorem proving community.

Language models have had some successful applications in the area of formal theorem proving [Polu and Sutskever, 2020, Han et al., 2021, Jiang et al., 2021, Polu et al., 2022]. However, we observe that language-model-based reasoning systems are inept at premise selection. The difficulty of premise selection for language models is that they cannot effectively reason over thousands of available facts

36th Conference on Neural Information Processing Systems (NeurIPS 2022).

and their definitions in plain text form. In Section 2.2, we elaborate on the scale of the problems language models need to deal with for premise selection and provide empirical evidence for this difficulty. Seeing that hammers are very good at finding relevant facts, we propose in our framework to offload the premise selection task from language models to hammers. The resulting system is Thor, a framework that organically integrates language models and ATPs via the use of hammers.

The methodology of Thor is simple and can be deployed in any hammer-enabled ITP: we first use the hammer method to attempt to prove every proof state in the training problems, and mark the successful application steps. Then we train the language model on the training problems, predicting a special token (e.g., `<hammer>`) if the hammer can be applied at the state. When doing evaluation, if the language model emits the special token, we invoke the hammer method. This methodology incurs very little extra computation compared to standard language model training while capitalising on the potential of a hybrid neuro-symbolic model.

To demonstrate the usefulness of Thor, we instantiate it with a language-model-based reasoning system for the ITP Isabelle and its implementation of the hammer method called Sledgehammer [Paulson, 2010]. We then investigate the performance of the instantiated Thor system on two datasets, PISA [Jiang et al., 2021] and MiniF2F [Zheng et al., 2022]. On PISA we dramatically improve the success rate of a language-model-based reasoning system from $39.0\%$ to $57.0\%$ and solve $8.2\%$ of problems that cannot be solved by either language models or Sledgehammer alone. On MiniF2F, Polu et al. [2022] used expert iteration to improve on a language model and achieved the state-of-the-art 1-pass success rate of $29.6\%$. With much less computation, Thor increases this rate to $29.9\%$, slightly surpassing the previous result. It is worth noting that Thor and expert iteration can be used in tandem.

In this paper, we demonstrate that finding suitable sub-systems for premise selection can benefit the performance of the overall reasoning system. Given Thor's strong performance, computational efficiency, and applicability to many ITPs, we believe it should become a strong baseline as often as possible when language models are used for theorem proving.

**Contributions**

1. We created Thor, a theorem proving framework which integrates language models and automated theorem provers via the use of hammers.

2. We raised the state-of-the-art success rate of language-model-based reasoning systems on PISA from $39.0\%$ to $57.0\%$. Thor proved $8.2\%$ theorems which cannot be proved by either language models or Sledgehammer.

3. We improved the state-of-the-art success rate on MiniF2F from $29.6\%$ to $29.9\%$, matching the language models trained with expert iteration, but with far less computation.

## 2 Background

### 2.1 Automated and Interactive Theorem Proving

Mechanising theorem proving has been a grand challenge of artificial intelligence since the late 1950s [Gelernter, 1959]. A group of systems, called automated theorem provers, attempt to use automated procedures to determine the validity of conjectures (e.g., the DPLL algorithm [Davis et al., 1962] for SAT problems [Tarski, 1969]). Popular examples of ATPs include E, SPASS, Z3, CVC4, and Vampire. Although SAT is known to be NP-complete [Cook, 1971], modern ATPs can often solve problems with millions of symbols [Ohrimenko et al., 2009] and are useful practically.

ATPs are often based on fragments of first-order logic, which limits the type of theorems they can express. Hence, projects such as the formalisation of complicated mathematical results [Gonthier et al., 2008, Avigad et al., 2007, Gonthier et al., 2013, Scholze, 2021] and operating system kernel verification [Klein et al., 2009] are done in interactive theorem provers, often based on higher-order logic or dependent type theory. ITPs and ATPs have very different objectives: ITPs aim at making it easy to formalise a diverse set of problems in numerous mathematical domains in a sound manner; while ATPs focus on improving the efficiency and performance on very well defined problems like SAT solving. Prominent ITPs include Isabelle, Mizar, HOL Light, HOL4, Lean, and Coq. Theorem proving in ITPs can be modelled as a sequential decision process: initially a theorem gets declared and the `proof state` contains some goals; at each step, the user produces a `proof step` that

applies to and transforms the `proof state`; when all the goals have been discharged, the theorem is considered proven. Large libraries of mathematical knowledge such as the Archive of Formal Proofs[1] and the Mizar Mathematical Library[2] have been built around these ITPs. Because of the size of these mathematical libraries, to find useful premises in them is a difficult problem. In the next subsections we illustrate how two different approaches deal with premise selection.

## 2.2 Language Models for Theorem Proving

Language models that automate theorem proving mostly follow the approach of the *GPT-f* model [Polu and Sutskever, 2020]: pre-trained causal language models are used to predict a `proof step` that can be applied, given the current `proof state` and some optional `context`. Concretely, a language model can take as input and output, two sequences of the following form:

```
INPUT:    <SOS> <CTXT> $(context) <PRF_STT> $(proof state) <PRF_STP>
OUTPUT:   $(proof step) <EOS>
```

At test time, the reasoning system receives the text representation of the current `proof state`, samples a `proof step` from the language model, applies it to the ITP, and repeats until the proof is finished or a computational budget has been reached. A best-first strategy is often used for proof search: a queue of search nodes is maintained with the priority being the accumulated log likelihood of the generated `proof steps`.

Language models treat all input and output information as text and they are usually limited to be a few thousands of characters long. To do premise selection well, the language model has to either memorise all the relevant premises during training, or be prompted with available premises in evaluation. It is difficult to do the former because a mathematical corpus can have too many facts for a language model to remember. For example, the Archive of Formal Proofs has more than 200,000 theorems, plus the numerous definitions and their derivations to serve as premises. The latter is no easier because there may be too many premises to fit into the input. For instance, if we use the textual representation of 300 available premises (a usual number used for premise selection with symbolic tools) and their definitions as the `context` in the input-output format above, the input length can well exceed 10,000 characters and the limit of standard language models. We observe that empirically 1.9% of the steps involving premise selection generated by the language model manage to advance the proof, while the number is 28.2% for steps having no premises. Hence, a good mechanism for premise selection could bring crucial benefits.

## 2.3 Hammers

Blanchette et al. [2016] define hammers as methods that "automate reasoning over large libraries developed with formal proof assistants (ITPs)". Consider, for example, Sledgehammer (designed for Isabelle) which is the original and the most popular implementation of hammers. Figure 1 presents a proof of $\sqrt{2} \notin \mathbb{Q}$ in Isabelle. The beauty of using Sledgehammer with Isabelle is that despite the complicated-looking proof, humans only need to sketch the proof in Figure 1a and let Sledgehammer find all the necessary premises to complete every single proof step. The final accepted proof is presented in Figure 1b. The Sledgehammer proof steps use the internal proof methods `metis`, `meson`, `smt`, `auto`, `simp`, `fastforce` and `blast`. Conveniently, this tells us which steps in the corpus are generated by Sledgehammer. Note that a human user might also use the proof methods `auto`, `simp`, `fastforce` and `blast` as these do not contain additional premises. Only the methods `metis`, `meson`, `smt` are exclusive to Sledgehammer.

We now describe how Sledgehammer performs premise selection: Sledgehammer makes it possible to leverage the advancement of ATP research while using ITPs, and can thus be seen as a bridge between the two [Paulson, 2010]. When invoked, Sledgehammer translates the current goal together with hundreds of possibly relevant premises into a format (e.g., SMT-LIB, TPTP) that external ATPs can understand [Meng and Paulson, 2008]. The ATPs are then executed to solve the current goal. Note that Isabelle follows a kernel philosophy (i.e., only a handful of axioms and inference rules are trusted), and external ATPs are used skeptically—should a proof be found by the ATPs, Sledgehammer picks out the useful premises, and reconstructs the proof within the Isabelle kernel

---

[1]https://www.isa-afp.org
[2]http://mizar.org/library/

```
lemma "sqrt 2 ∉ ℚ"
proof
  assume "sqrt 2 ∈ ℚ"
  then obtain a b::int where "sqrt 2 = a/b"
    "coprime a b" "b ≠ 0" sledgehammer
  then have c: "2 = a^2 / b^2"
    sledgehammer
  then have "b^2 ≠ 0" sledgehammer
  then have *: "2*b^2 = a^2"
    sledgehammer
  then have "even a"
    sledgehammer
  then obtain c::int where "a=2*c"
    sledgehammer
  with * have "b^2 = 2*c^2"
    sledgehammer
  then have "even b"
    sledgehammer
  with ‹coprime a b› ‹even a› ‹even b›
    show False sledgehammer
qed
```

(a) The proof sketch produced by the human user. The `sledgehammer` command indicates that the human invokes the Sledgehammer method at that point.

```
lemma "sqrt 2 ∉ ℚ"
proof
  assume "sqrt 2 ∈ ℚ"
  then obtain a b::int where "sqrt 2 = a/b" "coprime a b" "b ≠ 0"
    by (metis Rats_cases' less_irrefl)
  then have c: "2 = a^2 / b^2"
    by (smt (z3) of_int_power power_divide real_sqrt_pow2)
  then have "b^2 ≠ 0" by fastforce
  then have *: "2*b^2 = a^2"
    by (smt (verit, ccfv_SIG) c comm_semiring_class.distrib
        eq_divide_eq_numeral(1) mult_cancel_right1 numeral_Bit0
        numeral_plus_numeral of_int_add of_int_power
        of_int_power_eq_of_int_cancel_iff one_plus_numeral)
  then have "even a"
    by (smt (z3) even_power oddE)
  then obtain c::int where "a=2*c" by blast
  with * have "b^2 = 2*c^2" by auto
  then have "even b"
    by (smt (z3) even_power oddE)
  with ‹coprime a b› ‹even a› ‹even b› show False by fastforce
qed
```

(b) The proof accepted by Isabelle. The steps containing `assume`, `obtain`, `have`, `show` are from the original human proof sketch. The steps containing `metis`, `smt`, `fastforce`, `blast`, `auto`, `fastforce` are completed by Sledgehammer.

Figure 1: A proof of $\sqrt{2} \notin \mathbb{Q}$, adapted from the original by Li et al. [2021] with consent.

(e.g., using the primitive inference rules). Here, external ATPs serve as relevance filters of premises rather than trusted oracles. Hammers implemented in other ITPs are largely similar.

## 3   Thor

In this section we introduce Thor, a framework integrating language models and automated theorem provers via the use of hammers. Thor is motivated by the difficulty for language models to do premise selection and the excellent performance of hammers for it: we should be able to drastically improve automation in theorem proving if we can take the best from both worlds.

Below we provide the protocol of adopting Thor for a hammer-enabled ITP. We first provide Thor's training data preprocessing procedure in Algorithm 1, and then look at a concrete example to demonstrate its use.

---

**Algorithm 1** Thor's training data preprocessing algorithm.

---

**Require:** Proof state `s`, hammer method `h`
  `INPUT = s.input`
  **if** `h(s)` → `success` **then**                   ▷ Hammer can be applied to the proof state
    `OUTPUT = <hammer> <EOS>`
  **else**                                        ▷ Hammer fails at the proof state
    `OUTPUT = s.output`
  **end if**
  **return** `(INPUT, OUTPUT)`

---

Now consider the situation in the proof of $\sqrt{2} \notin \mathbb{Q}$ (Figure 1) after the step `then have "even a"`: without Thor, it should produce the following datapoint:

```
INPUT:   <SOS> <CTXT> $(context) <PRF_STT> $(proof state) <PRF_STP>
OUTPUT:  by (smt (z3) even_power oddE) <EOS>
```

With Thor's preprocessing, we apply the hammer method to the proof state and find out that it can be done successfully. Hence, we keep the input the same and change the output to:

```
OUTPUT:   <hammer> <EOS>
```

If the hammer method cannot be applied, we leave the datapoint unchanged. We iterate over every datapoint in the training data and apply this preprocessing algorithm.

We hypothesise that being exposed to training data in this format, the language model is capable of learning a heuristic for *when* the hammer can be successfully invoked. At evaluation time, whenever the language model outputs the sequence `<hammer> <EOS>`, instead of employing it directly in the ITP, we call the hammer method. This effectively makes the hammer an invokable method for the language model. This protocol is straightforward to implement for hammer-enabled ITPs.

The only extra cost of deploying Thor is in the data preprocessing step. Multiplying the hammer time limit by the average number of problems submitted to the Archive of Formal Proofs in one year, we estimate that 7400 CPU hours per year are needed to preprocess one of the largest proof corpora available. This is a modest cost since the process only needs to be done once per dataset and the results can be shared. Better still, for some ITPs, the hammer method leaves a trace, greatly reducing the time needed to figure out which steps can be solved by hammers. For the ITP Coq, all steps containing the keyword `sauto` are generated by CoqHammer [Czajka and Kaliszyk, 2018]. For Isabelle, all steps containing the keywords `metis`, `meson`, `smt` are generated by Sledgehammer (described in Section 2.3). With these traces, deploying Thor on ITPs like Coq or Isabelle incurs little extra computational cost compared to training a standard language model.

## 4 Experiment

Our experiments are intended to answer the following research questions:

1. Can Thor prove theorems that cannot be proved by language models or automated theorem provers individually? Does Thor improve premise selection for language models?

2. Does explicitly learning *how* to select premises hurt the performance of language models?

3. How important are the context information and the diversity of sequence generation?

4. How does Thor compare with other methods at improving language models for theorem proving?

To answer these questions, we create an instance of Thor for the ITP Isabelle. We choose Isabelle for two reasons: (1) Isabelle's Sledgehammer is one of the most mature hammer methods among major ITPs, and may thus showcase Thor's full potential; and (2) Isabelle's Archive of Formal Proofs is one of the world's largest formal mathematical libraries, suitable for data-hungry methods like language models. We make explicit the details of our experimental setup next.

### 4.1 Experimental Setup

**Machine specification**   For pre-training, fine-tuning, and evaluation, we use a TPUVM with 8 cores from Google Cloud Platform. The Isabelle process has access to up to 32 CPU cores. We estimate that reproducing all the experiments in this paper requires a total of 1160 TPU hours.

**Language model architecture**   We use a decoder-only transformer [Vaswani et al., 2017] language model, adapting the setup, codebase, and hyperparameters from [Wang and Komatsuzaki, 2021]. The language model has 700M non-embedding parameters, with 24 layers, 24 attention heads, a hidden dimension of 1536, and a GPT-2 [Radford et al., 2019] tokenizer with a vocabulary size of 50400. Rotary positional embeddings [Su et al., 2021] are used. The model is pre-trained on the GitHub + arXiv subsets of The Pile [Gao et al., 2021], with a context length of 2048. We use a global batch size of 32 sequences which amounts to 65536 tokens. For the first 3,000 steps, the learning rate linearly increases from 0 to 0.0002, and then it follows a cosine schedule with a final value of $1.2 \times 10^{-5}$ for 197,000 steps. We use a weight decay rate of 0.05 and no dropout for pre-training. Pre-training takes $\approx 150$ TPU hours. For fine-tuning, we use the procedure described in Section 3 to prepare the PISA dataset. We use the most recent `proof step` as the `context` in each datapoint. The same learning rate scheduling strategy is used, with a peak learning rate of $3 \times 10^{-4}$ after 10,000 steps and a final learning rate of $3 \times 10^{-5}$ after a further 90,000 steps. We use a dropout rate of 0.15 and a weight decay rate of 0.1. The global batch size is 256 sequences, or $524,288$ tokens. We early-stop fine-tuning and take the checkpoint at 11,000 steps for evaluation as the validation loss reaches a minimum then. Fine-tuning takes $\approx 50$ TPU hours.

**Sledgehammer configuration**    To set up Sledgehammer, we mostly follow the default Isabelle2021 configuration. An important default parameter is that the Sledgehammer timeout limit is 30s. Our configuration uses the on-machine versions of the five default ATPs (E, SPASS, Vampire, Z3, and CVC4) to prevent performance deviation caused by network issues.

**Proof search**    To sample from the language model, we use temperature sampling with the temperature parameter $T = 1.2$. To search for the proof of a theorem, we use the best-first search strategy described in [Polu and Sutskever, 2020]. The queue is ordered by the accumulated log likelihoods of the generated `proof steps`, with a maximum length of 32. Each `proof step` has a timeout limit of 10s. The search is terminated if and only if one of the following scenarios happens: (1) a valid proof has been found for the theorem; (2) the language model is queried 300 times; (3) a wallclock timeout of 500s has been reached; (4) the queue is empty but the theorem is not proved. Empirically, it takes $\approx 60$ TPU hours to evaluate $1,000$ problems.

Our language model setup is different from Language models of ISAbelle proofs [Jiang et al., 2021, LISA] in three aspects: (1) our language model has 700M instead of 163M non-embedding parameters (2) the most recent `proof step` is included in the language model prompt (3) a higher sampling temperature (1.2 instead of 1.0) is used.

## 4.2    Datasets and Environment

We use two datasets. The first is the PISA dataset [Jiang et al., 2021], which includes the Isabelle/HOL repository[3] under a BSD-style license and the Archive of Formal Proofs version 2021-10-22[4], whose various entries are under open-source licenses as described on its official page. PISA contains the core higher-order logic library of Isabelle, as well as a diverse library of proofs formalised with Isabelle, mostly concerning mathematics or verification of software and hardware. The PISA dataset contains 2.49 million datapoints in total. The `proof states` have an average length of 369 characters and the `proof steps` have an average length of 33 characters. All of the Isabelle/HOL theorems go into the training set as they are considered foundational and might be used by all other repositories. We make a $95\%/1\%/4\%$ split of theorems from the AFP for the training/validation/test sets. We randomly select 3,000 theorems from the test set (*PISA/test*) for the evaluation of model performance.

The second is the Isabelle fraction of the MiniF2F dataset [Zheng et al., 2022] under an Apache license. The dataset contains 488 high school mathematics competition problems split into a validation set and a test set, each with 244 problems. These problems have been formalised in Lean, Metamath, and Isabelle to provide a benchmark of the same problems in different ITP languages. This allows us to contrast different approaches developed for different ITPs. Since we do not use the validation set for model selection, we do not actually distinguish between the two sets. Hence, we mainly compare with previous work on the test set as the final result.

We use the codebase by Jiang et al. [2021], under a BSD 3-clause license, to interact with the Isabelle server and prove theorems from both datasets.

## 4.3    Thor Against an Ensemble of a Language Model and Sledgehammer

Because Thor has both a language model and Sledgehammer at its disposal, we wish to investigate how it fares against a simple ensemble of the two. We set out to evaluate the performance of Thor, as well as a language model of the same configuration, and Sledgehammer with a 120s timeout on *PISA/test*. It takes $\approx 50$ TPU hours to evaluate Thor for $1000$ problems. The proof success rates on *PISA/test* are presented in the second column of Table 1. We can see that the language model alone and Sledgehammer alone can prove $39.0\%$ and $25.7\%$ of the problems respectively. When we take the union of the problems they manage to solve individually, we get a $48.8\%$ success rate. Thor manages to prove $57.0\%$ of the problems. This implies that for $8.2\%$ of the problems, Thor uses both the language model and Sledgehammer to complete the proofs, and it is not possible to achieve this with only the language model or only Sledgehammer. We perform 4 case studies on problems that only Thor can solve in Appendix A.

---

[3]https://isabelle.in.tum.de/website-Isabelle2021/dist/library/HOL/index.html
[4]https://www.isa-afp.org/release/afp-2021-10-22.tar.gz

Table 1: Proof success rates on *PISA/test*

| Method | Success rate (%) |
|---|---|
| LISA [Jiang et al., 2021] | 33.2 |
| Sledgehammer | 25.7 |
| Language model | 39.0 |
| Language model ∪ Sledgehammer | 48.8 |
| Thor | **57.0** |

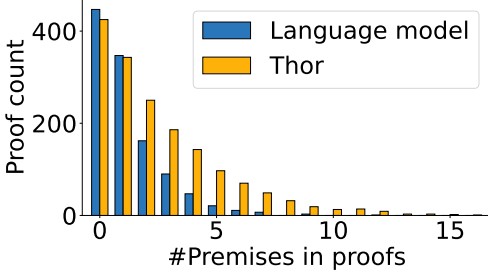 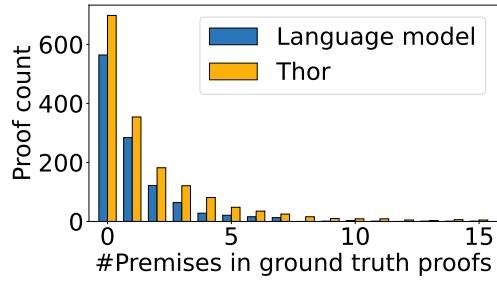

(a) The number of premises in successful proofs found by the language model and Thor.

(b) The number of premises in ground truth proofs for problems solved by the language model and Thor.

Figure 2: Comparison of the number of premises in problems the language model and Thor can solve.

Thor's motivation is to solve the premise selection problem for language models. To confirm that Thor helps premise selection, we collect the proofs generated by the language model and Thor respectively and count the number of premises in them. The results are presented in Figure 2a: we can see that for proofs requiring 0 or 1 premises, Thor and the language model perform similarly. But for proofs requiring more premises, Thor performs much more robustly, finding several times more proofs than the language model. We also count the number of premises in the ground truth proofs (written by humans) for theorems the language model and Thor can prove. The results are presented in Figure 2b: we see that whatever the number of premises the ground truth uses, Thor outperforms the language model in finding proofs, and the more premises the ground truth proof has, the more obvious is the effect. We conclude that Thor is indeed more capable of premise selection than language models.

## 4.4 The Effect of Learning When to Select Premises

We perform a brief statistical analysis to demonstrate the necessity of letting the language model learn when Sledgehammer shall be applied. In the training data, Sledgehammer is the output for $2.40\%$ of the steps. During evaluation, Sledgehammer execution accounts for $1.29\%$ of the proof steps. Of the $1.29\%$ Sledgehammer calls, $28.3\%$ are successful. Recall in Section 2.2, we observe that a vanilla language model succeeds at premise selection only $1.9\%$ of the time. We conclude that Thor learns from the training data to be highly cautious and effective with respect to executing Sledgehammer.

To empirically demonstrate the effect of learning when premise selection should be performed, we run an experiment with the same setup as Section 4.3. But instead of letting the language model decide when Sledgehammer shall be called, we execute Sledgehammer for every new proof state with a $30s$ timeout. This experiment ablates the effect of letting the language model learn *when* to call Sledgehammer completely, which resulted in a success rate of $51.7\%$ on the same test suite, $5.3\%$ lower than Thor. This success rate is reported in Table 2. Sledgehammer has a timeout of 30s, while an average step execution in Isabelle takes 7ms. The execution of 32 steps in a batch takes $210ms$, 136x faster than one Sledgehammer execution. Therefore, given a limited amount of time, the frequent querying of Sledgehammer severely limits the proof search nodes the model can explore. Hence it is detrimental to the model performance.

Additionally, the procedure we described in Section 3 ensures that the language model learns *when* to do premise selection, but not *how* to do it, by replacing the premise selection steps with `<hammer>`. Here we investigate the effect of making the language model learn both *when* and *how*. An easy way

Table 2: Proof success rates on *PISA/test*

| Variants of Thor | Success rate (%) |
| --- | --- |
| Base, sampling temperature $T = 1.2$ | 57.0 |
| Not learning *when* to select premises | 51.7 |
| Learning both *when* and *how* to select premises | 55.4 |
| No proof context | 53.6 |
| Sampling temperature $T = 1.0$ | 55.7 |

to achieve this is to create a variant of Thor: (i) at training time, use the original data; (ii) at evaluation time, when the language model outputs a sequence containing any of the Sledgehammer keywords, invoke Sledgehammer. This further simplifies data preparation and explicitly subjects the language model to perform premise selection. To investigate the effect of this alternative approach, we evaluate a system trained in this way on *PISA/test* and present its success rate in Table 2. We can see that it achieves a success rate of $55.4\%$ on *PISA/test*, $1.6\%$ lower than the base version of Thor, which suggests that explicitly learning *how* to do premise selection marginally decreases its success rate. This result is expected: since finding *how* to do premise selection is entrusted to the hammer method, the language model should focus on learning *when* to invoke the hammer for optimal performance. Making the language model learn an irrelevant additional task only hurts Thor's performance.

### 4.5 The Effect of the Proof Context

Our language model setup differs from that of LISA [Jiang et al., 2021] in that we use the most recent `proof step` as the `context` in the input data, as introduced in Section 3. This is based on the intuition that the most recent `proof step` information is beneficial for the language model's reasoning ability. In this subsection we perform an ablation study to confirm the effect of this `context` on Thor. Here a variant of Thor is trained without the `context` information and evaluated on *PISA/test*. The results are in Table 2. We observe that this variant manages to prove $53.6\%$ of theorems on *PISA/test*, $3.4\%$ fewer than the base version of Thor. The drop in success rate indicates that the `context` information we use is crucial for the optimal performance of Thor.

### 4.6 The Effect of the Sequence Sampling Diversity

Our language model setup differs from LISA [Jiang et al., 2021] also in the sampling temperature. Previous works on language models for theorem proving often use a temperature $T = 1.0$ [Polu and Sutskever, 2020, Jiang et al., 2021] for sampling output sequences, while we use $T = 1.2$. A higher temperature in the sampling procedure means that the generated sequences are more diverse (having a higher entropy). Here we perform an ablation study on the diversity of Thor-generated sequences. We evaluate Thor with sampling temperature $T = 1.0$ on *PISA/test* and the success rate is in Table 2. We can see that the success rate with sampling temperature $T = 1.0$ is $55.7\%$, $1.3\%$ lower than with $T = 1.2$. This suggests a more diverse sampling strategy can improve Thor's performance, and that the optimal diversity in language model samples varies for different systems.

### 4.7 Comparing Thor with Expert Iteration

There exist other methods for improving language models for theorem proving like value function training [Polu and Sutskever, 2020], proof artifact co-training [Han et al., 2021, PACT], and expert iteration [Polu et al., 2022]. We wish to compare Thor with them. However, these methods operate in ITPs other than Isabelle and are thus hard to compare with directly. Thankfully, Polu et al. [2022] used expert iteration [Silver et al., 2017] to improve PACT [Han et al., 2021] and to achieve the state-of-the-art result on MiniF2F, a dataset containing multiple ITP formalisations of the same problems. Hence, we can fairly contrast expert iteration with Thor. Expert iteration, as argued by Polu et al. [2022], provides an unsupervised curriculum for automated theorem provers: given suitable data, it provides samples similar to the test distribution for further training, thus narrowing the generalisation gap between training and test. We should emphasise that Thor and expert iteration are not incompatible methods: one can use Thor *together with* expert iteration.

Table 3: Proof success rates on *MiniF2F*.

| Method | Valid (%) | Test (%) |
|---|---|---|
| PACT [Han et al., 2021] | 23.9 | 24.6 |
| Expert iteration [Polu et al., 2022] | **33.6** | 29.6 |
| Sledgehammer | 9.9 | 10.4 |
| Language model | 25.0 | 24.2 |
| Language model $\cup$ Sledgehammer | 27.1 | 27.5 |
| Thor | 28.3 | **29.9** |

We start by evaluating Thor, a language model with the same configuration, and Sledgehammer on MiniF2F. The results are presented in Table 3. We also include the success rates of the language model that Polu et al. [2022] used (PACT), as well as the language model after expert iteration in the same table. The success rates on the validation set are also included, but we use the rates on the test set as the final results, as the valid set can be used for model selection. We can see that the language model is able to prove 24.2% of the problems on MiniF2F, similar to PACT's 24.6%. Thor increases the success rate of the language model by 5.7% to 29.9%, while expert iteration increases the success rate of PACT by 5.0% to 29.6%. Hence, the improvement in proof success rate brought upon the language model by Thor is comparable to that by expert iteration.

An important factor in choosing a suitable method is its cost. Expert iteration requires manually creating a set of "curriculum" problems, evaluating the language model on them, and training the language model on a growing training set for one epoch every iteration. We estimate that performing expert iteration on the same scale as Polu et al. [2022] for Isabelle, it would cost 100 human hours to formalise 300 maths problems, and 500 TPU hours to evaluate and fine-tune the language model for 8 expert iterations. Thor, on the other hand, incurs little extra computational cost compared with training a standard language model. We conclude that while requiring a much smaller computational budget, Thor can improve language models' success rates to a similar degree as expert iteration.

## 5   Related Work

Language models were first applied to automate theorem proving by Polu and Sutskever [2020]. Since then, there have been a few works [Han et al., 2021, Jiang et al., 2021, Polu et al., 2022] aiming to enhance the ability of language-model-based reasoning systems, or to enable these systems for interactive theorem provers that were not supported before. All of these works used the same framework as laid down by Polu and Sutskever [2020], namely to iteratively sample from a language model and directly apply the output to the ITP. Thor, to the best of our knowledge, is the first system to explicitly hybridise language models and symbolic reasoning tools (ATPs) for theorem proving. Instead of relying on language models entirely, Thor uses hammers, a well-established tool, to solve premise selection.

With the growing bodies of formal mathematical libraries, premise selection has become one of the most crucial tasks of theorem proving. The hammer method is one of the many ways that premise selection can be done. We have described how the Isabelle implementation of the hammer method selects premises in Section 2. HOL(y)Hammer [Kaliszyk and Urban, 2015] and CoqHammer [Czajka and Kaliszyk, 2018] implement the hammer method for HOL Light and Coq respectively, making it possible for Thor to be instantiated for them. Apart from hammers, SInE [Hoder and Voronkov, 2011] and SRASS [Sutcliffe and Puzis, 2007] are both symbolic methods that take on the task of premise selection by ranking the available premises according to their relevance to the current conjecture, measured by syntactic and semantic distances respectively. MaLARea [Urban, 2007] pioneered having machine learning components in premise selection systems and its later version MaLARea SG1 [Urban et al., 2008] combines machine learning and formal semantics for premise selection. A few approaches [Irving et al., 2016, Wang et al., 2017, Kaliszyk et al., 2017] use deep learning in the premise selection task. These diverse methods may have merits over the hammer approach, and thus have the potential to be integrated as the premise selection component for future versions of Thor.

# 6 Discussion

In this paper we introduced a simple approach to overcome language models' weakness in premise selection for theorem proving: we created Thor, a framework that integrates language models and automated theorem provers via the hammer proof method. We presented a straightforward protocol for deploying Thor on any hammer-enabled ITP (including Isabelle, HOL Light, Coq, etc.). The instance of Thor with Isabelle dramatically increased the number of automatically proved theorems, suggesting that language models' deficiency at premise selection can be effectively compensated by utilising ATPs. Furthermore, approaches like expert iteration [Polu et al., 2022] or proof artifact co-training [Han et al., 2021] have no contradictions and can be easily incorporated with Thor. Compared with these methods, Thor has the additional advantage of being computationally efficient.

One limitation of Thor is that it only admits automated theorem provers that directly generate valid proof steps in the ITP via the use of the hammer. In Section 5, we pointed out that there are other premise selection tools with approaches different from the hammer method that the current version of Thor cannot use. Also, there exist methods which assist premise selection but do not directly generate the proof steps. An example of this is SErAPIS [Stathopoulos et al., 2020], which performs semantic search over the Isabelle mathematical library with the help of Wikipedia. Thor cannot use this class of methods either. We leave to future work the task of broadening the options for the premise selection tool that Thor uses. Here we only tested Thor on the ITP Isabelle due to the computational costs of experiments. Therefore another future direction is to instantiate Thor with other ITPs and see whether improvements brought by Thor are as significant for other ITPs as we show them here for Isabelle.

Thor demonstrates how a difficult problem for language models can be solved by borrowing tools from another research domain. We are encouraged by its success and think more problems like premise selection can be solved similarly. For example, Collins et al. [2022] demonstrated how a symbolic planner can make a language model more robust. With its strong performance, computational efficiency, and convenient deployment, Thor gives scope to tool hybridisation, which shows promise to be impactful in the field of automated reasoning, and artificial intelligence in general.

## Acknowledgement

We thank Yiannos Stathopoulos and Lawrence Paulson for helpful discussions. AQJ is supported by a Peterhouse Graduate Research Studentship. WL is supported by the ERC Advanced Grant ALEXANDRIA (Project GA 742178). MJ is supported by the EPSRC grant EP/T019603/1.

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
