# OpenReview forum: "Thor: Wielding Hammers to Integrate Language Models and Automated Theorem Provers"
_NeurIPS.cc/2022/Conference — NeurIPS 2022 Accept_

### Official Review · Reviewer_vtgR · 2022-07-11

**Rating:** 8
**Confidence:** 3
**Soundness:** 3 good
**Presentation:** 4 excellent
**Contribution:** 3 good

**Summary:**

The paper presents a method to integrate language models and hammers (Automated
Theorem Provers) for Interactive Theorem Proving. The authors train the language
model to recognize an opportunity to invoke a hammer by transforming the
training data: they check whether the hammer can be applied at each proof state,
and if it can, then they change the next proof step the model should output to
<hammer>. A language model trained on this data can thus output either the next
proof step, or <hammer> to indicate that a hammer should be invoked at that
proof step.



**Questions:**

Can you think of another domain where the idea of the paper could be applied?


**Limitations:**

The limitations are properly addressed.


**Strengths And Weaknesses:**

## Strengths

The core idea of the method is very simple and elegant, could be applied widely,
and it outperforms the state-of-the-art.

The paper is very clearly written and explains the concepts from theorem proving
which are needed to understand the paper.

There are some directions of future work mentioned which could improve the
system further (e.g., expert iteration).

## Weaknesses

One weakness of the method could be that it requires the processing of the whole
training corpus and applying the hammers, which could be costly.

---

> ### Author Response · Authors · 2022-08-01
> **Response to reviewer vtgR**
>
> We thank the reviewer for their comments, helpful suggestions, and encouragement. We are glad that the reviewer finds the contributions sound and the findings novel. Here we reply to the reviewer’s comments and questions:
>
> > **One weakness of the method could be that it requires the processing of the whole training corpus and applying the hammers, which could be costly.**
>
> We mentioned in section 3 that processing the whole training corpus needs to be done once per dataset and approximately 7400 CPU hours are needed per year for one of the largest proof corpora available. With high-performance computing, this is a moderate cost. However, we fully seize the reviewer’s concern that the computational cost might impede the community from performing research based on Thor. Hence, when the paper is released, we will accompany it with the archive of formal proofs corpus pre-processed for Thor, thus reducing the community of the reproduction and development cost. We will report the actual cost of doing so too.
>
> > **Can you think of another domain where the idea of the paper could be applied?**
>
> We think that the specific technique of learning and applying sledgehammer is limited to the domain of automated theorem proving: although we only demonstrated the use of Thor with Isabelle in the paper, Thor also applies to other interactive theorem provers (e.g., HOL Light, Coq, and HOL4) where hammers have been implemented.
>
> However, we consider the idea of the paper to be: for tasks difficult for language models, it is beneficial to learn how to invoke symbolic methods to solve them, when they exist. For example: [Collins et al., 2022] combined a large language model and a symbolic planner to solve the task of planning, which outperforms the large language model alone. We have updated Section 6 to include the two examples given.
>
> References:
> Collins, K. M., Wong, C., Feng, J., Wei, M., & Tenenbaum, J. B. (2022). Structured, flexible, and robust: benchmarking and improving large language models towards more human-like behavior in out-of-distribution reasoning tasks. arXiv preprint arXiv:2205.05718.

---

> > ### Comment · Reviewer_vtgR · 2022-08-07
> > **Thank you for your thoughtful answers**
> >
> > Thank you. Preprocessing the archive of proofs will be very helpful for future research and it will be also interesting to see how much computational power is required to do that. Also thank you for the interesting reference.

---

### Official Review · Reviewer_uriH · 2022-07-14

**Rating:** 6
**Confidence:** 4
**Soundness:** 3 good
**Presentation:** 3 good
**Contribution:** 2 fair

**Summary:**

This paper presents Thor, a new learning-based theorem prover in the Isabelle interactive theorem prover. Thor combines the neural-based search heuristics with the Sledgehammer hammer in Isabelle. A language model is trained to generate the next proof steps, and Sledgehammer is one of the actions the model could output. This new prover achieves the new state-of-the-art result on the PISA dataset and performs comparatively with the best existing methods on the MiniF2F dataset.

=========================================
The authors' responses addressed my concerns about if it is necessary to learn to call the hammer. I increased my rating to weak accept.

**Questions:**

How often are Sledgehammer called by the language models? How often do the Sledgehammer calls solve the goals?

**Limitations:**

I didn't find specific limitations of this paper. The proposed approach is general and can be simply combined with many neural theorem provers.

**Strengths And Weaknesses:**

Strengths
1 It is a reasonable idea that we can improve the neural theorem provers with the hammers, considering that hammers would call the automated theorem provers that are designed and tuned carefully by the experts.
2 Experiments show that Thor has significant advantages over the provers that are based on language models or hammers only.

Weakness
The approach to combining the hammers with the language models is fairly simple and straightforward. I doubt if it is necessary to have the language model learn to call Sledgehammer. Can we simply call Sledgehammer on each new proof state we want to expand during the proof search? What would be the performance of this simple baseline?

---

> ### Author Response · Authors · 2022-08-01
> **Response to reviewer uriH**
>
> We thank the reviewer for their comments and helpful suggestions. Here we first address the reviewer’s comment on the weakness of the model and then answer the reviewer’s question.
>
> > **Weakness The approach to combining the hammers with the language models is fairly simple and straightforward. I doubt if it is necessary to have the language model learn to call Sledgehammer. Can we simply call Sledgehammer on each new proof state we want to expand during the proof search? What would be the performance of this simple baseline?**
>
> The reviewer points out a very reasonable missing baseline. We greatly appreciate the suggestion and add the baseline experiment. However, on whether it helps to have the language model learn when sledgehammer should be called, we respectfully disagree. To support our stance, we first state a fact regarding the speed of executing sledgehammer vs an average step, then contrast our method with the baseline analytically, and finally give the performance of the baseline.
>
> Speed comparison fact: The Sledgehammer setup in Isabelle is optimised for a 30s timeout, meaning that the probability of sledgehammer finding a solution drastically decreases after 30s of running. This makes it reasonable to use 30s as the timeout for sledgehammer calls. In contrast, a candidate step proposed by the language model, that does not involve sledgehammer, takes 7ms to execute on average. If we execute 32 candidate steps, it will take 220ms on average. There is a 136x time difference between running and not running sledgehammer in a single step.
>
> Analytic comparison: We agree with the reviewer that calling sledgehammer on each new proof state will only improve the performance of the model, given infinite time. However, in time-limited evaluations (the case for automated theorem proving works and competitions), the difference in execution time is crucial. Letting the language model learn when to execute sledgehammer allows time savings on states where sledgehammer is unlikely to succeed (e.g. requiring induction).
>
> We ran a baseline experiment that the reviewer suggested, calling sledgehammer on each new proof state. This resulted in a success rate of 51.7% on the same dataset. 5.3% lower than the result with Thor. Therefore we think it a necessity for our model to discern when sledgehammer should be executed.
>
> > **How often are Sledgehammer called by the language models? How often do the Sledgehammer calls solve the goals?**
>
> With the model that has the highest success rate on the AFP test set, we present the following statistics: 1.29% of the model’s step executions made during evaluation on the test set are sledgehammer calls. 28.3% of sledgehammer calls succeed to solve the given sub-goal. We see that sledgehammer is called sparingly so as to not waste too much time on its execution.
>
> We have added a new Appendix B (We would have preferred to put it in the main paper for the final version. It is added as an appendix due to the page limit constraint) to give statistics about sledgehammer, and introduce the results from the baseline experiment. We further explain why the baseline has an inferior performance. We hope the additional content makes the answers to your questions clear from the paper.

---

> > ### Author Response · Authors · 2022-08-09
> > **Does our rebuttal address your concerns? Please consider increasing your score.**
> >
> > Does the rebuttal we provided address your concern regarding the baseline and do the statistics convince you that learning to use hammer is essential? If so, please consider increasing your score. If not, can you please provide further feedback on weaknesses so we can improve? Many thanks!

---

### Official Review · Reviewer_AFQP · 2022-07-25

**Rating:** 8
**Confidence:** 4
**Soundness:** 4 excellent
**Presentation:** 4 excellent
**Contribution:** 3 good

**Summary:**

Language models (LMs) have difficulty selecting the correct premise for a proof when the library of available premises is large. The authors use a data pre-processing step to train an LM to use an automatic theorem prover to improve premise selection. Combining the best of both worlds (LM theorem proving and automatic theorem proving), they achieve new state of the art performance on the PISA dataset and computationally efficient methods for matching best methods for MiniF2F.

**Questions:**

Expert iteration (EI) is able to achieve similar performance to Thor, but this paper fails to give an explanation for why.
At one point in the paper (line 104), you are able to identify empirically that premise selection is a major bottleneck in LM-based theorem proving, and that Thor improves LMs by improving premise selection. Does EI also improve premise selection? If not, do you know why it improves LM performance? If EI improves LM performance in another way, is there an estimation for how much combined Thor and EI would improve performance?

**Limitations:**

The authors primarily give methodological limitations (rather than discussing negative societal impacts). This seems reasonable given the domain (automatic theorem proving) primarily impacts the mathematical research community and is less of a "real world" application.

**Strengths And Weaknesses:**

Originality
This paper claims to be the first to use hammers for LM-based automatic theorem proving. A logical strategy to use, but nonetheless (as far as this reviewer is aware) novel.

Quality
The paper is thorough in its ablations, and presents a compelling case for their findings.

Clarity
The general writing style of this paper is straightforward and clear.
The paper could be more clear about the fact that their technique is primarily a data-preprocessing step that allows a language model to take advantage of an ATP, rather than a novel architecture.

Significance
Improved results on LM-based theorem proving is an exciting new development.

---

> ### Author Response · Authors · 2022-08-01
> **Response to reviewer AFQP**
>
> We thank the reviewer for their comments, helpful suggestions, and encouragement. We are glad that the reviewer finds the contributions sound and the findings sound. Here we answer the reviewer’s questions:
>
> > **Expert iteration (EI) is able to achieve … Does EI also improve premise selection? If not, do you know why it improves LM performance? If EI improves LM performance in another way, is there an estimation for how much combined Thor and EI would improve performance?**
>
> These are reasonable questions, but we wish to first emphasise that expert iteration (EI) is not something we propose in the paper, and EI can be used in tandem with Thor. Hence we think the discussion around expert iteration is connected with the essence of the paper.
>
> We think that the principal reason for EI’s effectiveness for theorem proving is that it narrows the generalisation gap between training and test data: in the work of [Polu et al., 2022], the training data is mainly the mathematical library of Lean, which consists of advanced mathematical constructions while the test data is the miniF2F dataset, which consists of high-school mathematical competition problems. Hence in order to narrow the generalisation gap, they introduced a curriculum whose distribution is similar to the test. We do not think EI will be particularly helpful with premise selection, for the following reason: EI can help boost certain proof patterns if the model is able to successfully apply them on the curriculum. We argued in the paper that premise selection is difficult because the huge volumes of premises may fall outside of the context windows of language models so they will struggle to find useful premises on the curriculum in the first place. Therefore EI will not boost language models’ premise selection abilities.
>
> We have a concurrent submission (attached as a supplementary material) in which we combine Thor and EI. We refer the reviewer to table 3 of that paper for how much the combination improves the performance on the MiniF2F dataset: We showed that two expert iterations on the MATH dataset problems increase the performance of Thor by 9% and 5.3% respectively on the validation and test partitions of MiniF2F. We think this empirical evidence demonstrates that EI might improve the model in other ways than selecting premises.
>
> In summary, the addition of EI can help improve the performance of Thor, but we do not think it is because of further improving premise selection, as argued above. We have updated Section 4.7, the discussion on EI, to make our standpoint clear from the paper.
>
> References:
> Polu, S., Han, J. M., Zheng, K., Baksys, M., Babuschkin, I., & Sutskever, I. (2022). Formal mathematics statement curriculum learning. arXiv preprint arXiv:2202.01344.

---

### Author Response · Authors · 2022-08-01
**General comment to all reviewers, ACs, and SACs**

We would like to first thank all the reviewers, area chairs, and senior area chairs for their generous inputs and efforts for this paper. We are very lucky to receive your feedback, which we treasure and utilise to improve the paper.

We have updated the paper according to the reviews. The difference in the paper is highlighted in brown. Due to the page limit constraint during the rebuttal phase, we are unable to add too much additional content to the main paper. We hope the elaborated rebuttals, together with the paper update, illustrates our stance clearly. Moreover, if the paper were accepted, we would utilise the additional page allowance to make sure the points raised are properly addressed in the paper.

We also updated the supplementary material to include a concurrent submission of ours, to provide further evidence to some points we made during the rebuttal.

We would love to have more exchanges with the reviewers, ACs, and SACs during the later phases, and thanks again for your time and energy!

---

### Meta-Review · Area_Chair_DVHv · 2022-08-27

**Recommendation:** Accept
**Confidence:** Certain

**Metareview:**

The paper presents a method to integrate language models and hammers (Automated Theorem Provers) for Interactive Theorem Proving. The authors train the language model to recognize an opportunity to invoke a hammer by transforming the training data: they check whether the hammer can be applied at each proof state.

The approach is novel, while being simple. The reviewers are generally happy with the writing of the work. There were some concerns (such as obvious baselines, pointed out byuriH) that seem to be mitigated.
vtgR pointed out the slowness of preprocessing, which I think is an issue that should be explicitly acknowledged in the revision and addressed in future work.

Given the overall positive feedback of the reviewers, I am recommending an "accept" for this work.

**Award:**

No

---

### Decision · Program_Chairs · 2022-09-14

Accept